# Identification of potential biomarkers in dengue via integrated bioinformatic analysis

Li-Min Xie[1,2☯], Xin Yin[1,3☯], Jie Bi[4☯], Huan-Min Luo[1,2], Xun-Jie Cao[1,2], Yu-Wen Ma[1,2], Ye-Ling Liu[1,2], Jian-Wen Su[1,2], Geng-Ling Lin[1,2], Xu-Guang Guo[1,2,5,6]*

**1** Department of Clinical Laboratory Medicine, The Third Affiliated Hospital of Guangzhou Medical University, Guangzhou, China, **2** Department of Clinical Medicine, The Third Clinical School of Guangzhou Medical University, Guangzhou, China, **3** Department of Pediatrics, The pediatrics school of Guangzhou Medical University, Guangzhou, China, **4** School of Food Science and Engineering, Wuhan Polytechnic University, Wuhan, China, **5** Key Laboratory for Major Obstetric Diseases of Guangdong Province, The Third Affiliated Hospital of Guangzhou Medical University, Guangzhou, China, **6** Key Laboratory of Reproduction and Genetics of Guangdong Higher Education Institutes, The Third Affiliated Hospital of Guangzhou Medical University, Guangzhou, China

☯ These authors contributed equally to this work.
* gysygxg@gmail.com

**Data Availability Statement:** The authors confirm that all data underlying the findings are fully available without restriction. All gene files are available from the Gene Expression Omnibus

## Abstract

Dengue fever virus (DENV) is a global health threat that is becoming increasingly critical. However, the pathogenesis of dengue has not yet been fully elucidated. In this study, we employed bioinformatics analysis to identify potential biomarkers related to dengue fever and clarify their underlying mechanisms. The results showed that there were 668, 1901, and 8283 differentially expressed genes between the dengue-infected samples and normal samples in the GSE28405, GSE38246, and GSE51808 datasets, respectively. Through overlapping, a total of 69 differentially expressed genes (DEGs) were identified, of which 51 were upregulated and 18 were downregulated. We identified twelve hub genes, including MX1, IFI44L, IFI44, IFI27, ISG15, STAT1, IFI35, OAS3, OAS2, OAS1, IFI6, and USP18. Except for IFI44 and STAT1, the others were statistically significant after validation. We predicted the related microRNAs (miRNAs) of these 12 target genes through the database miRTar-Base, and finally obtained one important miRNA: has-mir-146a-5p. In addition, gene ontology (GO) and the Kyoto Encyclopedia of Genes and Genomes (KEGG) pathway enrichment were carried out, and a protein–protein interaction (PPI) network was constructed to gain insight into the actions of DEGs. In conclusion, our study displayed the effectiveness of bioinformatics analysis methods in screening potential pathogenic genes in dengue fever and their underlying mechanisms. Further, we successfully predicted IFI44L and IFI6, as potential biomarkers with DENV infection, providing promising targets for the treatment of dengue fever to a certain extent.

## Author summary

Dengue fever is a mosquito borne viral disease caused by a single stranded RNA virus with four serotypes. DENV infection can cause various diseases, such as breakbone fever,

database (accession numbers: GSE28405, GSE38246, GSE51808 and GSE84331).

**Funding:** The author(s) received no specific funding for this work.

**Competing interests:** The authors have declared that no competing interests exist.

haemorrhagic fever, and shock syndrome. As one of the most viral diseases leading to incidence rate and mortality in animal arthropods, Dengue fever has become an increasingly serious global health threat. However, the pathogenesis of dengue fever has not been fully elucidated. In this study, we used bioinformatics analysis to identify potential biomarkers associated with dengue fever and elucidate their underlying mechanisms. Finally, we predicted that IFI44L and IFI6 might be potential biomarkers of DENV infection. This finding provides a promising target for the treatment of dengue fever to a certain extent. In addition, the Gene Ontology (GO), Kyoto Encyclopedia of Genes and Genomes (KEGG) pathway enrichment, protein–protein interaction (PPI) network were implemented to analyze the key differentially expressed genes after DENV infection, and the related mechanisms were illuminated by this study.

## Introduction

In the tropical and subtropical parts of the world, dengue fever virus (DENV) infection has become an increasingly common health concern. Due to the large geographic extent, increase in the number of cases, and severity of the disease, the DENV infection has evolved from a sporadic disease to a major public health problem with significant social and economic impacts [1–4]. Dengue is a mosquitoes-transmitted viral disease caused by a single-stranded RNA virus, which has four serotypes (DENV 1–4)[5]. DENV infection can cause various illnesses, such as breakbone fever, haemorrhagic fever, and shock syndrome[6]. Dengue divides into three phases: the febrile phase with acute onset of fever, the critical phase with metabolic acidosis and severe haemorrhage, and the recovery phase with resolved symptoms[3]. At present, some clinical trials have been conducted to reduce the effects and symptoms of dengue[7–9]. There are a few dengue vaccines but no specific antiviral treatment[3,5]. A DENV vaccine cannot elicit protection in naive individuals but only those with prior exposure, in addition, that is not equally protective against all four serotypes[10].

As one of the most viral diseases transmitted by arthropods that causes human morbidity and mortality, numerous studies have been performed to explore the pathogenesis of disease; The mainstream view is that immunity leads to cytokine storm, which leads to vascular leak and thus contributes to severe dengue disease in secondary infections[11]. However, many patients with DENV infection do not develop plasma leakage[12]. Plasma leak typically occurs in the critical phase[13], which is at the end of the acute phase[14]. Therefore, the febrile phase with or without the critical phase is the acute phase, which may lead to severe dengue[14]. A hypothesis based on molecular mimicry posits that some DENV-induced antibodies can cross-react with host proteins. A study verifies that the level of pre-existing anti-DENV antibodies is directly associated with the severity of secondary dengue disease in humans[11]. In sum, it still remains unclear, more research is needed to understand the potential pathogenesis in dengue.

In view of heterogeneity, biomarkers for reliably predict the development of severe dengue among symptomatic individuals are desperately needed in current research. The currently utilized warning signs to predict severe dengue are based on clinical parameters that appear late in the disease course and are neither sensitive nor specific. It promotes not only continued morbidity and mortality, but also ineffective patient triage and resource allocation[15]. Provided that we have had highly discriminating biomarkers, then developed a single, robust clinical algorithm, it will be broadly applicable across all age groups and in different locations[16],

which is meaningful to predict severe dengue and differentiate dengue-infected diseases with similar clinical phenotypes.

Microarray data analysis can identify DEGs in dengue fever patients with differing disease severity[2]. In addition, an increasing amount of evidence indicates the potential role of micro-RNAs (miRNAs) in regulating DENV[17,18]. MiRNAs are small non-coding RNA molecules that can regulate gene expression by inhibiting messenger RNA (mRNA) translation or inducing mRNA degradation[17]. Recently, Pong et al. reported that, with a DENV-1 infection, 23 highly differentially expressed miRNAs jointly modulate the adaptive immune response involving TGF-β, MAPK, PI3K-Akt, Rap1, Wnt, and Ras signalling pathways[19].

In this study, we performed a biological information analysis using microarray data and identified the DEGs for the infected and normal samples. Subsequently, the Gene Ontology (GO), Kyoto Encyclopedia of Genes and Genomes (KEGG) pathway enrichment, protein–protein interaction (PPI) network, and miRNA-target gene interaction network were analysed to understand the molecular mechanisms underlying dengue fever. In conclusion, our study aimed to explore the molecular biomarkers of dengue based on bioinformatic analysis and provide candidate biomarkers for early diagnosis and therapeutic targets.

## Materials and methods

### Microarray data

The Gene Expression Omnibus is a public and functional genomics database that contains high throughput gene expression data, chips, and microarrays. In this study, the GSE28405 [20], GSE38246[21], and GSE51808[22] microarray data were downloaded for analysis. Considering that the transcriptional profiles between the fever phase and convalescence phase in dengue patients are are quite different, we only included the data of samples collected in fever patients and the control group from these three data sets. Additionally, GSE28405, GSE38246, and GSE51808 are consisting of 26, 8, and 9 control samples and 31,105 and 28 infected samples in fever, respectively.

### Data processing

We used the R 4.0.1 statistical software (https://www.r-project.org/) and a Bioconductor (http://bioconductor.org/biocLite.R) to process raw data and screen differentially expressed genes. The data of GSE28405 and GSE38246 were batch calibrated and standardized by using the *limma* package. *Limma* package contains particularly powerful tools for reading, standardizing and exploring such data, and its core component is to fit gene linear model to gene expression data to evaluate the ability of differential expression[23]. The data of GSE51808 was batch calibrated and standardized by using the *affy* package. The differentially expressed genes were then filtered using a *limma* package. The screening threshold was p-value < 0.05 and fold-change ≥ 1.5. The *ggplot2* package was used to visualise the DEGs into a volcano map, while the *pheatmap* package was used to cluster the significant DEGs.

### Function and pathway enrichment analysis of DEGs

The Gene Ontology (GO, http://www.geneontology.org) is a community-based bioinformatics resource. It provides information about genes and gene product functions and uses ontology to enhance biological knowledge[24]. The KEGG (https://www.kegg.jp/) is a database for the qualitative interpretation of genomic sequences and other biological data, including systematic, genomic, and chemical information as well as an additional human-specific category of health information[25]. The related biological functions and signal pathways were analysed

using GO/KEGG enrichment and analysed again with the cluster Profiler software package, with $p < 0.05$ considered to be statistically significant.

## PPI network construction and identification and validation of hub genes

Protein–protein interaction (PPI) network analysis plays a major role in predicting the function of interacting proteins. It is a feasible tool that can be used to understand cell function and disease mechanism[26,27]. The STRING database (http://string-db.org) focuses on providing a key assessment of protein–protein interactions by integrating a large number of known and predicted protein–protein association data[28,29]. A PPI network visualised by the Cytoscape software was constructed by using the STRING database. Furthermore, the cyto-hubba plug-in of Cytoscape software was used to analyze the interaction of proteins and screen out hub genes with a higher score in this analysis, which means that they have higher connectivity in PPI networks. The statistical significance of these genes was verified by GSE84331[30] microarray data analysed using GEO2R. GEO2R is an interactive network tool that allows users to compare two or more sets of samples in a GEO sequence to identify differentially expressed genes[31]. $P < 0.05$ was considered statistically significant.

## MiRNA-target gene network

MicroRNA (miRNA) is a type of small endogenous non-coding RNA with 18–25 nucleotides. It is the main central regulatory factor at the post-transcriptional levels. It is involved in many biological processes such as cell cycle, cell differentiation, and apoptosis, among others[32,33]. The miRTarBase database contains manually managed and experimentally validated miRNA–gene interactions as well as detailed metadata, experimental methods, and conditions[32]. Accordingly, we constructed the miRNA–gene targeting relationship for overlapping differential genes and hub genes based on the miRTarBase database.

# Results

## Identification of DEGs

The datasets included are shown in Table 1. After analysing the GSE28405 dataset, we screened 668 DEGs, including 364 upregulated genes and 304 downregulated genes (Fig 1A); GSE38246 and GSE51808 were used to screen 1901 DEGs (924 upregulated and 977 downregulated) and 8283 DEGs (4165 upregulated and 4118 downregulated) respectively (Fig 1C and 1E). After screening the differential genes based on the volcano map, cluster analysis was carried out, as shown in Fig 1B, 1D and 1F. Finally, through a Venn analysis, 69 common DEGs were identified from three datasets, including 51 upregulated genes and 18 downregulated genes, which were subsequently used for further study (Fig 2 and Table 2).

## GO and KEGG pathway analysis

The detailed results of the GO enrichment analysis and KEGG pathway analysis of GSE28405, GSE38246, and GSE51808 are shown in Figs 3, 4 and 5. The type I interferon signaling pathway, DNA replication, and chromosome segregation were relatively enriched from these data sets in biological processes. In terms of cell components, the results showed that cytosolic ribosome, organellar large ribosomal subunit, mitochondrial ribosome, and mitochondrial large ribosomal subunit were significantly enriched. Concerning molecular function, ATPase activity, DNA-dependent ATPase activity, single-stranded DNA helicase activity, and catalytic activity, acting on DNA play an important role in the enrichment results relatively. For KEGG

**Table 1. Details of the data sources from Gene Expression Omnibus(GEO) for this study.**

| Reference | GEO Series (GSE) | Sample | Sample size | Normal vs Infection | GEO Platform (GPL) |
|---|---|---|---|---|---|
| Tolfvenstam et al (2011) | GSE28405 | Whole blood | 57 | 26 vs 31 | GPL2700 Sentrix HumanRef-8 Expression BeadChip |
| Popper et al(2012) | GSE38246 | Peripheral blood mononuclear cell (PBMC) | 113 | 8 vs 105 | GPL15615 SMD Print_1430 hr1 |
| Kwissa et al(2014) | GSE51808 | Whole blood | 37 | 9 vs 28 | GPL13158 [HT_HG-U133_Plus_PM] Affymetrix HT HG-U133+ PM Array Plate |
| Chandele et al (2016) | GSE84331 | Peripheral blood mononuclear cell (PBMC) | 12 | 5 vs 7 | GPL570 [HG-U133_Plus_2] Affymetrix Human Genome U133 Plus 2.0 Array |

pathway enrichment analysis, the relatively enriched pathways were the coronavirus disease, cell cycle, DNA replication, and protein processing.

## Protein–protein interaction work of overlapped DEGs and identification and validation of key genes

To identify the potential interactions between overlapping DEGs, a PPI network was constructed on the STRING, consisting of 69 nodes (genes) and 174 edges (Fig 6A). The foremost module in the PPI network was identified by MCODE, and 12 genes were identified as hub genes (Fig 6B). After verifying with GSE84331, 10 genes (MX1, IFI44L, IFI44, IFI27, ISG15, STAT1, IFI35, OAS3, OAS2, OAS1, IFI6, USP18) were statistically significant (Fig 7). Among them, IFN inducible protein 44-like (IFI44L), and IFNα inducible protein 6 (IFI6) were found to have a higher score in the PPI network and a lower p-value in the analysis.

## MiRNA-target gene network

The networks of miRNA-gene targeting relationship of 69 overlapping DEGs and 12 hub genes based on miRTarBase database are respectively presented in Fig 8A and 8B. According to the miRNA interactions and number and importance of target genes, has-mir-146a-5p was attained.

## Discussion

DENV infection can further lead to recessive infection, dengue fever, and severe dengue fever [34,35]. It is mainly transmitted to humans through female Aedes mosquitoes. Aedes mosquitoes are widespread in the tropical and subtropical regions of the world, putting nearly two-thirds of the world's population at risk[36,37]. Therefore, screening the potential biomarkers or exploring the related mechanisms through bioinformatics may contribute to the efficient diagnosis and treatment of dengue fever.

GO analysis can annotate genes and gene products involving cellular components, biological processes, and molecular functions[38]. In biological processes, DEGs are most enriched in type I interferon signaling pathway, DNA replication, and chromosome segregation. Dengue virus infection activates the innate immune system of the body to increase the secretion of interferon. Type I interferon signal transduction can fight against a variety of viruses that invade the human body. DENV can antagonize its signal transduction and promote its genome replication in host cells[39]. The imbalance of the cell cycle and mitotic cycle after DENV infection will affect DNA replication and cell proliferation. In terms of cell components, DEGs are significantly enriched in the cytosolic ribosome, organellar large ribosomal

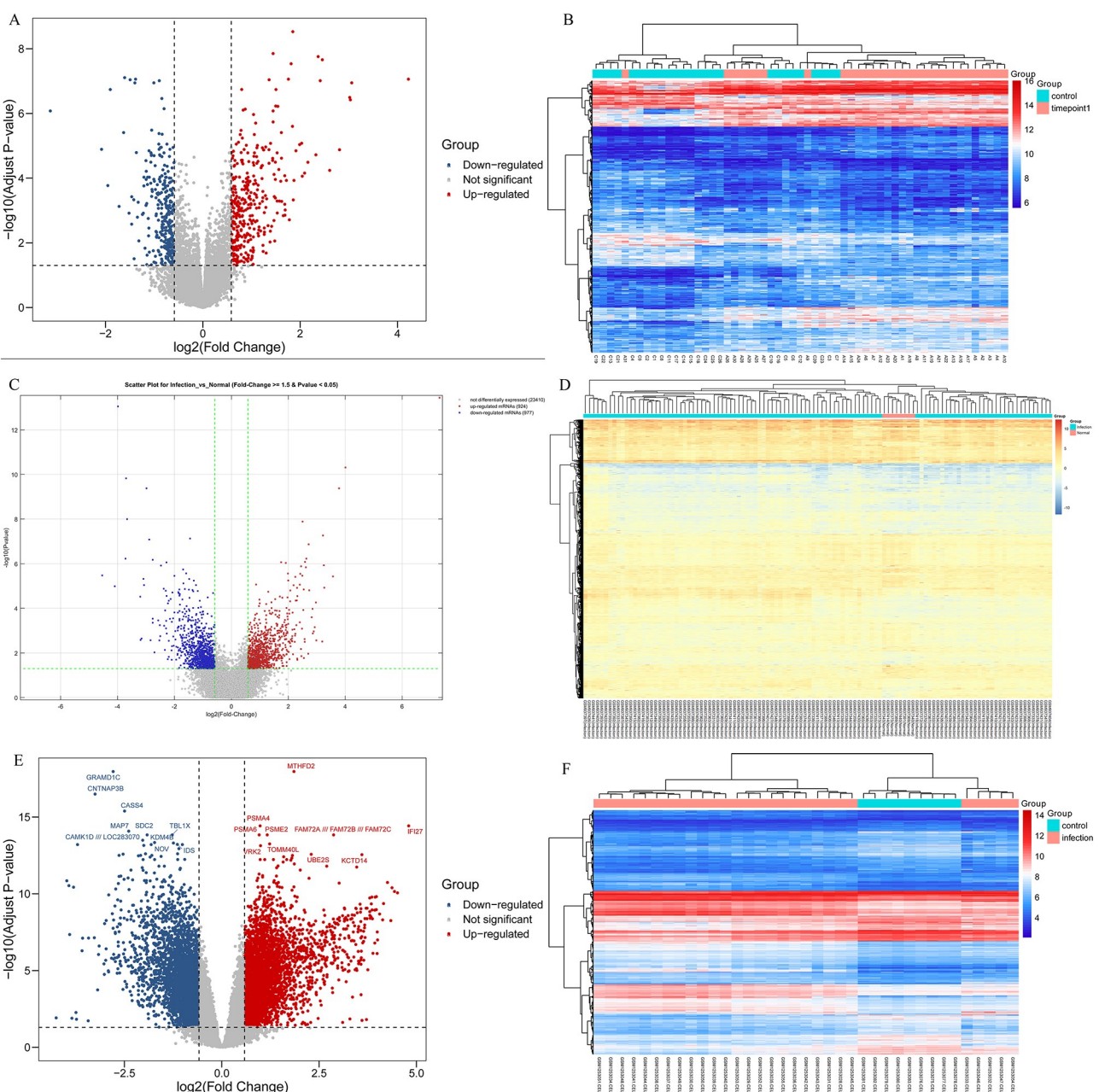

**Fig 1. Volcano map and heat map of differentially expressed genes (DEGs) in GSE28405(AB), GSE38246(CD), and GSE51808(EF).** (ACE) Red dots indicated up-regulated genes and blue dots indicated down-regulated genes. Black dots indicated the rest of the genes with no significant expression change. The threshold was set as followed: P<0.05 and |log2FC|≥2. FC: fold change. (BDF) Gene expression data is converted into a data matrix. Each column represents the genetic data of a sample, and each row represents a gene. The color of each cell represents the expression level, and there are references to expression levels in different colors in the upper right corner of the figure.

subunit, mitochondrial ribosome, and mitochondrial large ribosomal subunit, which is related to the enrichment of DNA replication and chromosome segregation in biological processes. Dengue virus genome replication in the cytoplasm of the host cell and take advantage of the host cell organelles to protein synthesis and assembly[40]. The protein is synthesized in the

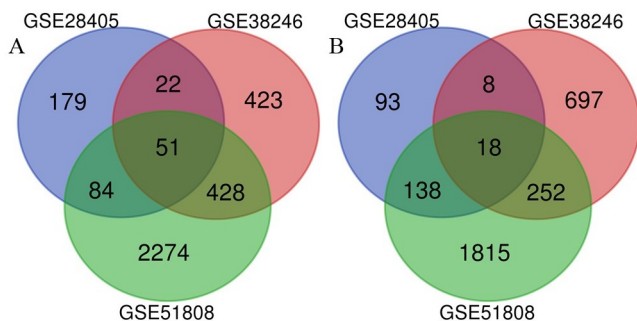

**Fig 2. The intersection results of GSE28405, GSE38246, and GSE51808.**

cytosolic ribosome, binding to a receptor protein such as the TOM complex on the outer membrane of the mitochondria, and is introduced into the mitochondria in an unfolded conformation, where it is eventually folded and assembled into the intrinsic structure[41]. In terms of molecular function, ATPase activity, DNA-dependent ATPase activity, single-stranded DNA helicase activity, and catalytic activity, acting on DNA play an important role in the enrichment results. ATP activation increases, generating cyclic adenosine phosphate (cAMP) under the action of adenylate cyclase. cAMP may trigger the fusion of secretory vesicles, again may by increasing vesicles and plasma membrane fusion between the diameter of the hole and opening time to adjust have fusion of secretory vesicles[42]. In this case, the diameter of the fusion hole and open time increasing, may increasing the dengue virus E protein involved in virus and nuclear fusion peptide in the somatic cell membrane fusion process, and promoting DENV through holes to promote infection in genetic material into cells.

Coronavirus disease 2019 (COVID-19) was found to be associated with dengue fever. It is hypothesized that dengue fever and COVID-19 share the same pathophysiological pathway, resulting in plasma leakage, thrombocytopenia, and coagulopathy are the hallmarks they both have[43]. Failing to diagnose dengue fever because of a false-positive test result for confirmed COVID-19, so we speculate that antibody cross-reactivity may exist in serology tests[44,45].

**Table 2. Up-regulated genes and down-regulated genes of overlapping DEGs.**

| Overlapping DEGs | | Gene terms |
|---|---|---|
| All | 69 | GOSR2, STAT1, MRPL17, THAP8, NR1H3, CBR1, MRPS18C, TOR3A, NAPA, BAK1, HIST1H4H, SIL1, BST2, TRIP6, C1QC, HIST1H2BD, DNASE2, C2, MAGED2, ISG20, SIGLEC1, IFI27L1, IFI35, TNNT1, SCO2, EPHB2, ATF5, CFB, OAS1, MT1F, OAS2, CTSD, IFI44, IFI6, HESX1, CD38, FDXR, MX1, KCTD14, C1QB, OAS3, LAG3, IFI44L, LGALS3BP, ISG15, CXCL10, LY6E, SPATS2L, TCN2, USP18, IFI27, CAMK1D, CMTM2, VENTX, FRY, ZFP36L2, SORL1, THBD, KLRB1, ITPKB, CIITA, CD22, MS4A1, LYST, CXCR5, PTGS2, STMN3, IVNS1ABP, TMEM71 |
| Up-regulated | 51 | GOSR2, STAT1, MRPL17, THAP8, NR1H3, CBR1, MRPS18C, TOR3A, NAPA, BAK1, HIST1H4H, SIL1, BST2, TRIP6, C1QC, HIST1H2BD, DNASE2, C2, MAGED2, ISG20, SIGLEC1, IFI27L1, IFI35, TNNT1, SCO2, EPHB2, ATF5, CFB, OAS1, MT1F, OAS2, CTSD, IFI44, IFI6, HESX1, CD38, FDXR, MX1, KCTD14, C1QB, OAS3, LAG3, IFI44L, LGALS3BP, ISG15, CXCL10, LY6E, SPATS2L, TCN2, USP18, IFI27 |
| Down-regulated | 18 | CAMK1D, CMTM2, VENTX, FRY, ZFP36L2, SORL1, THBD, KLRB1, ITPKB, CIITA, CD22, MS4A1, LYST, CXCR5, PTGS2, STMN3, IVNS1ABP, TMEM71 |

Abbreviation: DEGs, differentially expressed genes.

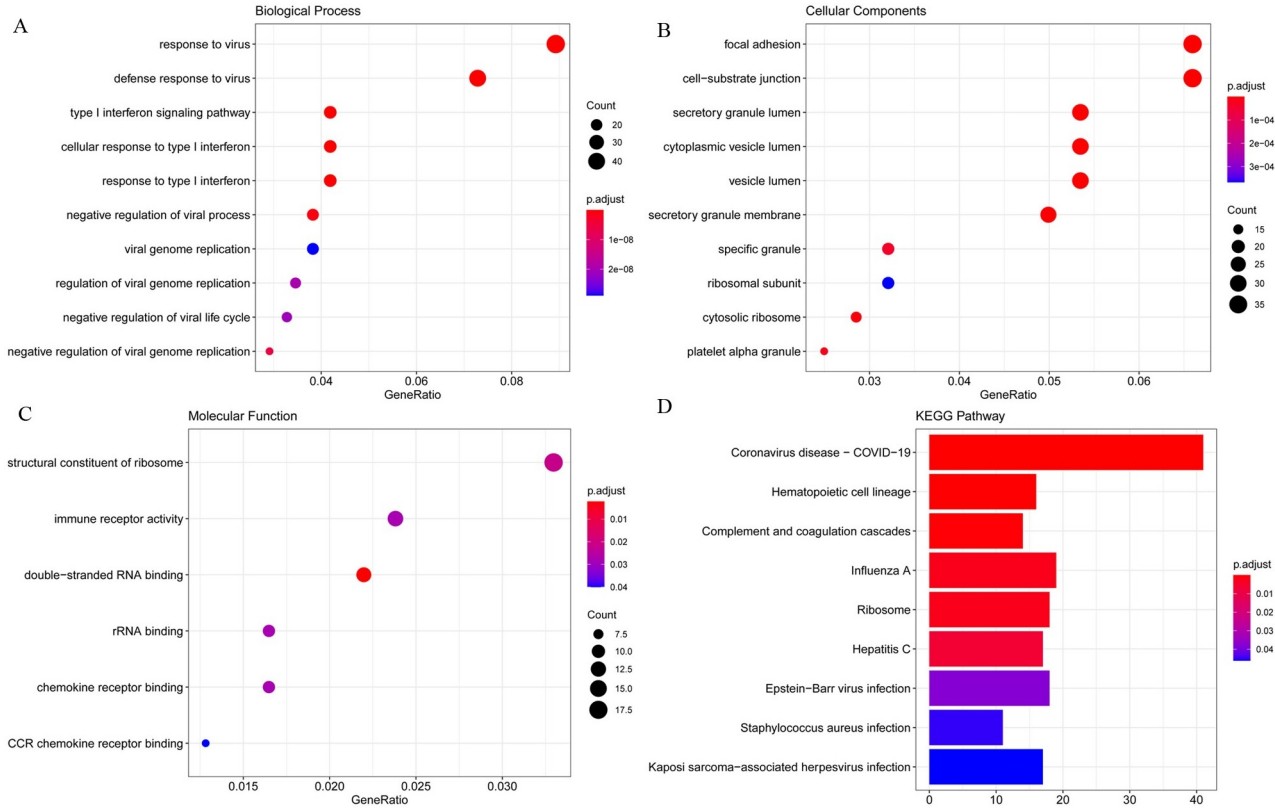

**Fig 3. The GO enrichment analysis and KEGG pathways analysis of GSE28405.** Abbreviation: GO, Gene Ontology; KEGG, Kyoto Encyclopedia of Genes and Genomes; DEGs, differentially expressed genes.

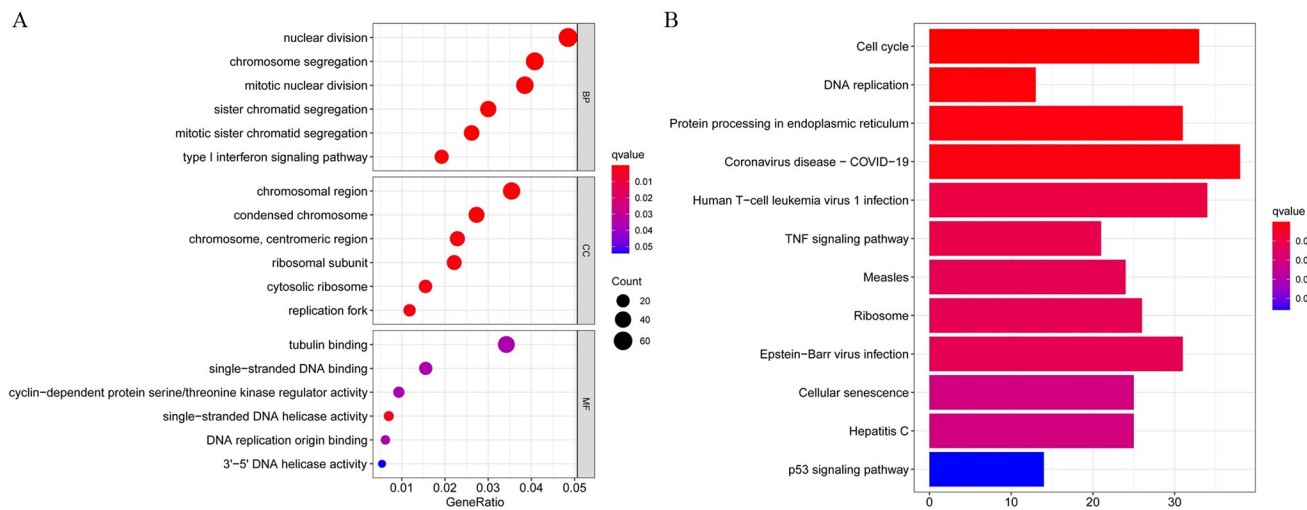

**Fig 4. The GO enrichment analysis and KEGG pathways analysis of GSE38246.** Abbreviation: GO, Gene Ontology; KEGG, Kyoto Encyclopedia of Genes and Genomes; DEGs, differentially expressed genes.

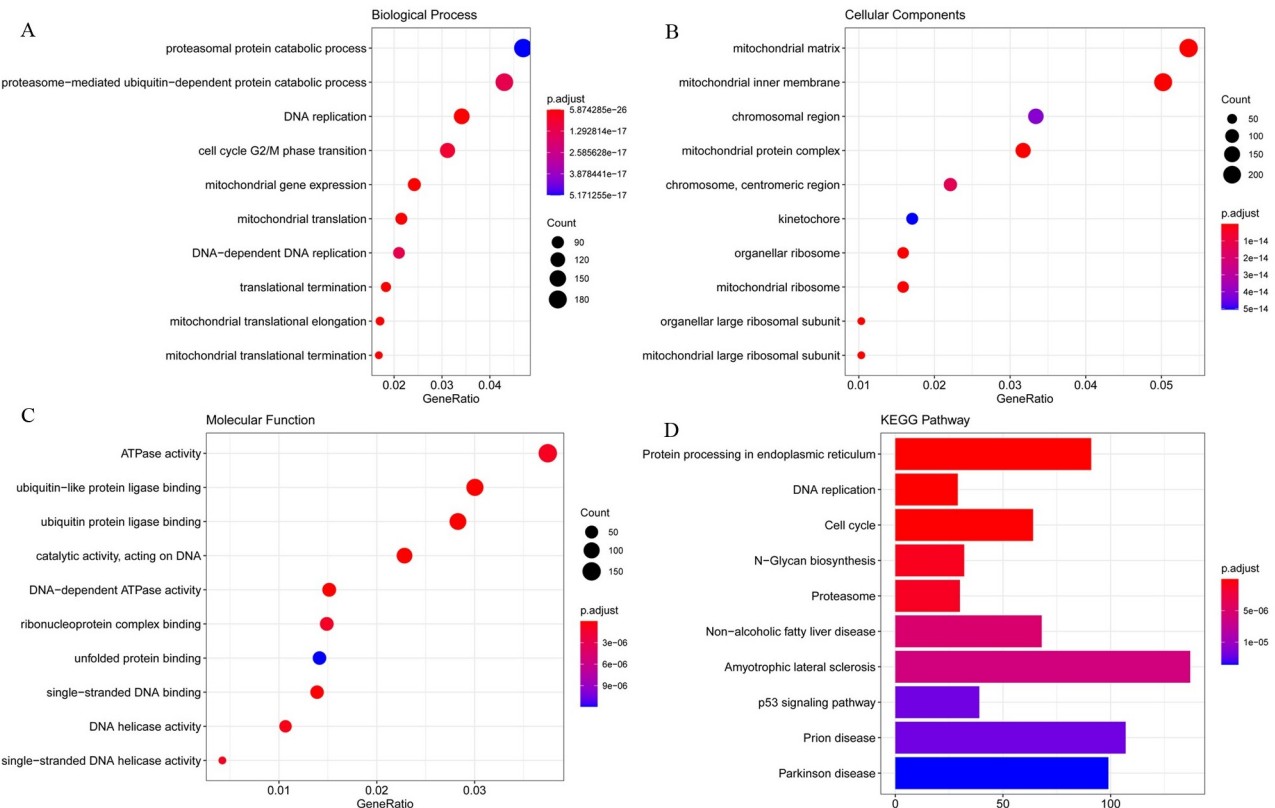

**Fig 5. The GO enrichment analysis and KEGG pathways analysis of GSE51808.** Abbreviation: GO, Gene Ontology; KEGG, Kyoto Encyclopedia of Genes and Genomes; DEGs, differentially expressed genes.

Cell cycle, DNA replication, and protein processing were significantly enriched in KEGG pathway enrichment analysis, all of which are closely related to cell division. Knockdown of cyclin-dependent kinase 8/19-cyclinC (Cdk8/19-cyclin C) reduced genome replication of the dengue virus and mitochondrial function in infected and uninfected cells as well as weakened glucose

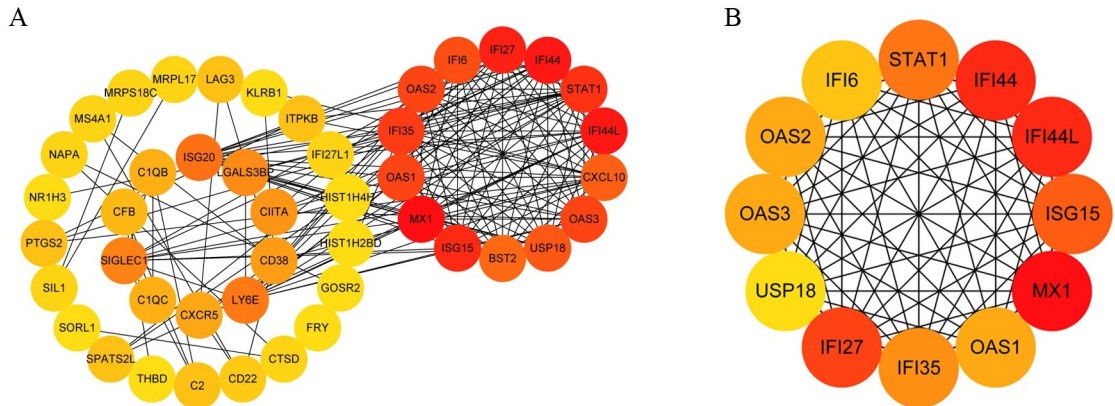

**Fig 6. The PPI network of overlapping DEGs (A) and the the important module of PPI network (B).** Abbreviation: PPI, protein–protein interaction; DEGs, differentially expressed genes.

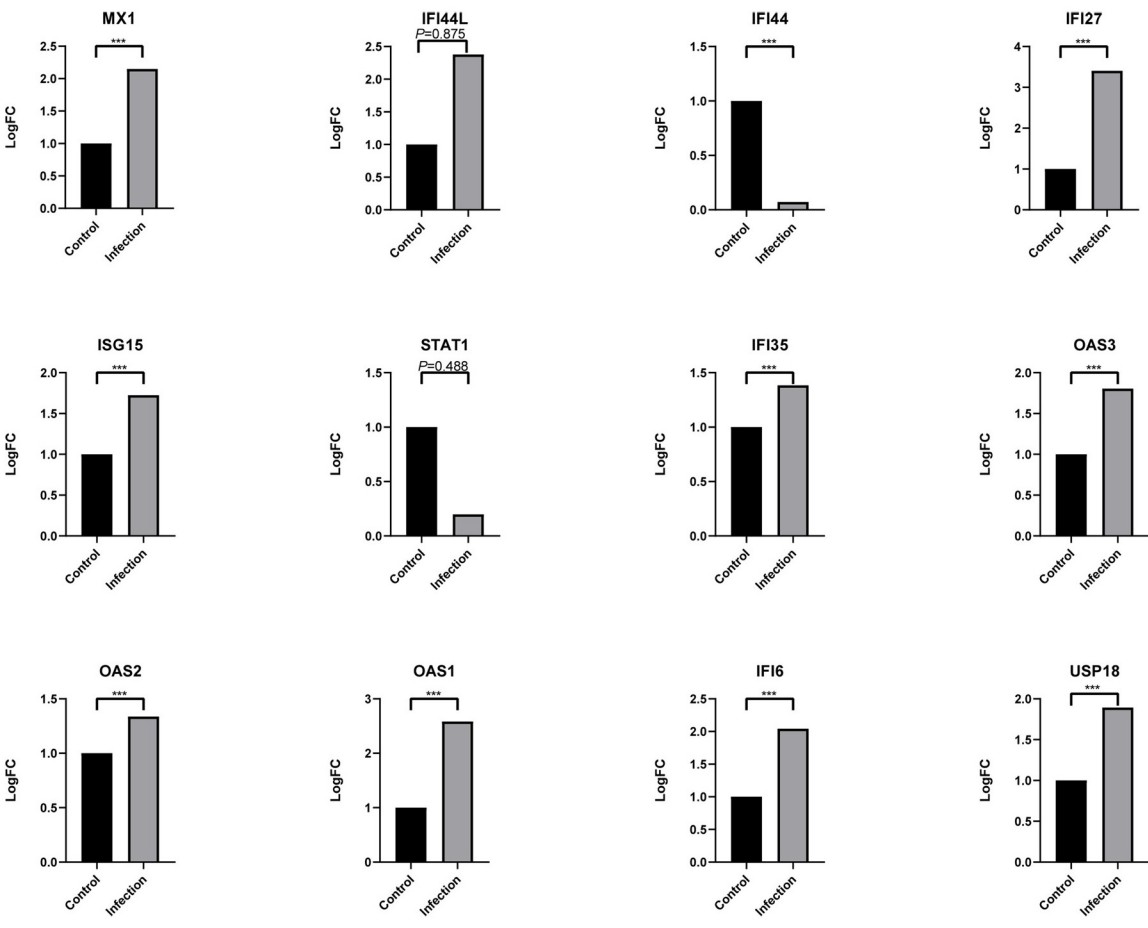

**Fig 7. Verification of hub genes.** *P*-value < 0.01 is considered to be statistically significant. (\*\*\**P*<0.001).

metabolism and autophagy to inhibit viral replication and metabolism[46]. Cyclin G-associated kinases (GAK) phosphorylated adaptor protein complexes (APs), thereby regulating membrane transport and promoting the dengue virus infection[47]. GAK inhibitors and their derivatives showed antiviral activity against the dengue virus[48]. XuM et al. found that the

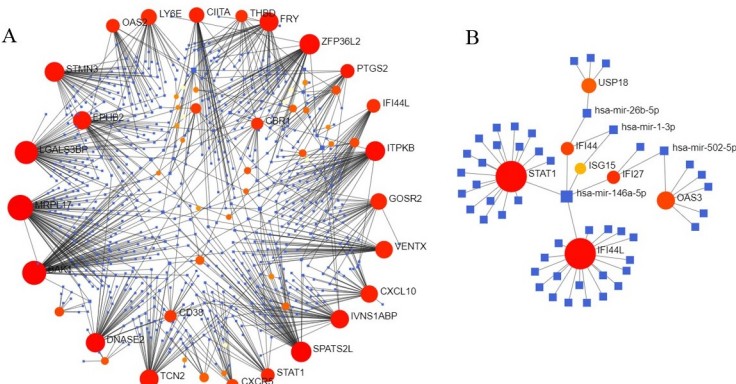

**Fig 8. The miRNA-target gene network of overlapping DEGs (A) and hub genes (B) based on miRTarBase v8.0 database.** Abbreviation: miRNA, microRNA, DEGs, differentially expressed genes.

infection of Zika virus (ZIKV), a Flaviviridae similar to the dengue virus, was closely related to cell cycle regulation; Hammack et al. found that ZIKV suspends host DNA replication during the S phase and induces DNA damage response and enhanced virus replication, which may occur with the dengue virus as well[49]. Therefore, cell division is a significant pathway for virus replication and infection, including cell cycle, DNA replication, and protein processing.

The PPI could help us understand protein–protein interactions; the rich interaction in gene expression of the dengue virus underscores the potential role of regulating host gene expression during infection[50]. Screening the most important module and its verification showed higher degrees of IFI44L and IFI6 in the PPI network and lower p-values in the analysis, which potentially indicates their significant association with dengue fever. Remarkably, DENV-infected germ cells upregulated IFI44L by 130-fold confirmed in qRT-PCR, but not in ZIKV-infected germ cells[51]. It is uncovered that IFI44L supports different viruses replication, and negatively regulating type I IFN response induced[52]. IFI44L contributes to DENV infection due to low levels of type I IFN response showed in patients with severe dengue disease[53,54]. The canonical pathway of Type I IFN is activated by the IFN-stimulated gene like IFI6, which is up-regulated DENV infection[55]. Overall, IFI6 was demonstrated a high level of protection against DENV infection, by inhibited DENV2-induced autophagy and apoptosis[55–57]. They influence the occurrence and development of dengue virus infection through a complex and undefined network of interactions. MiRNA is a small non-coding RNA molecule, and there is an increasing evidence that the imbalance of miRNA results in many diseases. The MiRTar-Base v8.0 database was used to predict miRNAs based on top the 12 hub genes, and miR-146a-5p was the core miRNA. miR-146a-5p, a type I IFN-mediated regulator targeting NF-kb, had a high correlation with the platelets and white blood cells count, especially in neutrophils and lymphocytes in initially diagnosed dengue fever[58,59]. Furthermore, activitied serum aspartate transaminase (AST) and alanine aminotransferase (ALT) also indicate miR-146a-5p affect liver complication in infected dengue[58]. Exosome miR-146a can act on different immune cells, making recipient cells more susceptible to many viral infections[60]. Surprisingly, some studies assessed that miR-146a-5p is associated with induced autophagy, which is a process in cell degradation and recycling for DENV replication[61–63]. Thus, miR-146a-5p has the potential to serve as a circulating biomarker for dengue pathogenesis.

This study has some limitations. Firstly, a small sample size would increase the error of research results to a certain extent. If the analysis can be based on large sample size, it is possible to more fully study the relationship between each central gene and pathway to improve the accuracy. Meanwhile, there are different samples in different data sets we included, such as whole blood and peripheral blood, which would cause some errors in our results. Finally, due to ethical issues and lack of funding, we did not conduct in vitro experiments to further verify our results but chose to use the results of another data set for validation analysis. This will also affect our conclusion to a certain extent.

Dengue is now the most important mosquito-borne disease after malaria and can cause serious complications such as bleeding or severe shock syndrome[64]. Therefore, it is urgent to study the pathogenesis of dengue fever. In conclusion, we have studied the microarray data of normal samples and DENV-infected samples through bioinformatics analysis to identify the differentially expressed genes after dengue infection. Another microarray data set was then used to verify genes of important modules, which resulted in 12 statistically significant hub genes. Their associated miRNAs were then predicted based on the miRTarBase database. Finally, we predicted that IFI44L, IFI6, and mir-146a-5p can be used as potential biomarkers of dengue infection, Our study may have potential implications for future prediction of disease progression in symptomatic dengue patients, and has important significance for the pathogenesis and targeted therapy of dengue.

## Acknowledgments

We acknowledge GEO database for providing their platforms and contributors for uploading their meaningful datasets.

## Author Contributions

**Conceptualization:** Xu-Guang Guo.

**Data curation:** Li-Min Xie.

**Formal analysis:** Li-Min Xie, Jie Bi, Xun-Jie Cao.

**Methodology:** Xu-Guang Guo.

**Project administration:** Xin Yin, Jie Bi, Xu-Guang Guo.

**Supervision:** Li-Min Xie, Jie Bi.

**Validation:** Li-Min Xie.

**Visualization:** Li-Min Xie, Xin Yin, Jie Bi, Huan-Min Luo, Xun-Jie Cao, Yu-Wen Ma.

**Writing – original draft:** Li-Min Xie, Xin Yin, Huan-Min Luo, Yu-Wen Ma, Ye-Ling Liu, Jian-Wen Su, Geng-Ling Lin.

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
