## [Decision Letter · Decision Letter 0]

9 Mar 2021

Dear Mr. Guo,

Thank you very much for submitting your manuscript "Identification of potential biomarkers in dengue via integrated bioinformatic analysis" for consideration at PLOS Neglected Tropical Diseases. As with all papers reviewed by the journal, your manuscript was reviewed by members of the editorial board and by several independent reviewers. In light of the reviews (below this email), we would like to invite the resubmission of a significantly-revised version that takes into account the reviewers' comments. 

We cannot make any decision about publication until we have seen the revised manuscript and your response to the reviewers' comments. Your revised manuscript is also likely to be sent to reviewers for further evaluation.

Sincerely,

Ahmed Mostafa

Associate Editor

A. Desiree LaBeaud

Deputy Editor

Reviewer's Responses to Questions

**Key Review Criteria Required for Acceptance?**

**Methods**

-Are the objectives of the study clearly articulated with a clear testable hypothesis stated?

-Is the study design appropriate to address the stated objectives?

-Is the population clearly described and appropriate for the hypothesis being tested?

-Is the sample size sufficient to ensure adequate power to address the hypothesis being tested?

-Were correct statistical analysis used to support conclusions?

-Are there concerns about ethical or regulatory requirements being met?

Reviewer #1: ALL CRITERIA ARE MET FOR METHODS

Reviewer #2: Are the objectives of the study clearly articulated with a clear testable hypothesis stated?

- The objective of the study is to identify dengue infection biomarkers. In order to do so the authors analyze public data with a standard methodology.

Is the study design appropriate to address the stated objectives?

- Yes, it is. By analyzing available transcriptomes, filtering DEGs, proceeding with pathway enrichment and then looking for miRNAs, the authors are able to find potential biomarkers for dengue infection. However, without at least an in vitro experiment in KO or of some sort targeted pathways in cell line DENV-infected x control in order to validate what was found, the findings in this publication are susceptible to criticism.

Is the population clearly described and appropriate for the hypothesis being tested?

- In fact, no. All three of the datasets chosen for this study have data from different timepoints of infection (post symptoms onset). During infections and diseases progression, patients’ transcriptional profile is in constant change and dengue infection is no exception. In fact, dengue infection is known to exhaust T and B lymphocytes (present in PBMCs, the samples from these studies) during initial stages of the infection but this changes in the convalescent stage. Since the transcriptional profile of each stage is so different, which of the samples were used and compared? There are no mentions to this point in the manuscript. 

Is the sample size sufficient to ensure adequate power to address the hypothesis being tested?

- Yes, the datasets combined have enough samples to ensure adequate power if all the samples are used. However, I do not believe a transcriptional profile from a convalescent patient would be equal from a patient in the initial stages of the infection. Since the study is trying to find biomarkers, by analyzing all of this data together the statistical tests should be removing potential DEGs from the initial stages just because they are not differentially expressed two weeks later. That being the case, should the authors take this into consideration, I cannot guarantee there are enough samples in these three datasets. 

Were correct statistical analysis used to support conclusions?

- Yes, both limma and GEO2R are bioinformatics’ tools that are already published and are highly accepted by scientists.

Are there concerns about ethical or regulatory requirements being met?

- No, there are not.

Reviewer #3: Yes the objectives are clearly stated, still few things are lacking in the manuscript.

Yes the sample size is good enough to address the hypothesis.

1) As all the three gene expression data come from same disease condition but other factors are highly varying. The authors should explain in detail the significance and preselection condition for these datasets to consider in the study.

2) The author should state which method were used for data normalization.

3) What is the rational for selection of fold change value?

**Results**

-Does the analysis presented match the analysis plan?

-Are the results clearly and completely presented?

-Are the figures (Tables, Images) of sufficient quality for clarity?

Reviewer #1: all criteria are met

Reviewer #2: Does the analysis presented match the analysis plan?

- Yes, it does.

Are the results clearly and completely presented?

- Yes, they are.

Are the figures (Tables, Images) of sufficient quality for clarity?

- Yes, they are.

Reviewer #3: 1) Yes, The analysis is presented matches the analysis plan

2) In results section 

Figure 5 is missing

Figure 5c is missing

Figure 6 an 7 description is missing in the text.

3) In discussion part references are missing.

The author either missed or overlooked some parts in the results and discussion, it should be carefully checked and edited.

**Conclusions**

-Are the conclusions supported by the data presented?

-Are the limitations of analysis clearly described?

-Do the authors discuss how these data can be helpful to advance our understanding of the topic under study?

-Is public health relevance addressed?

Reviewer #1: The conclusions are ok except for the microRNA part , which is not done correctly to predict microRNAs as biomarkers for deng.

Reviewer #2: Are the conclusions supported by the data presented?

- Yes

Are the limitations of analysis clearly described?

- No

Do the authors discuss how these data can be helpful to advance our understanding of the topic under study?

- Yes, both in the introduction and in the conclusion

Is public health relevance addressed?

- Yes, identifying potential biomarkers for disease treatment (and in this case, infection control) are of extreme importance. Even more when talking about dengue, its relevance goes without saying given the number of infected people yearly.

Reviewer #3: 1) Yes the conclusion is well supported by the data presented, limitation of the study is clearly stated.

2) The author should add few points relating to future use of outcome in the conclusion section.

3) The author should corelate the study with public health importance in the conclusion section

**Editorial and Data Presentation Modifications?**

Reviewer #1: it is ok

Reviewer #2: I strongly suggest an English revisor in case the paper is accepted due to some English mistakes. Text without numerated lines so it is hard to reference, but some examples in the introduction are:

"There is a few dengue vaccine but no specific antiviral treatment" -> are … vaccines

"A DENV vaccine can not elicits protection" -> cannot elicit

"mainstream view is that immunity leads to cytokine storm cytokine storm" -> repetition

"many patients with DENV infection without developing plasma leakage" -> do not develop

"which is meaningful to predict sever dengue" -> severe

Also, there is an inconsistency (probably a typo in the abstract?):

"Through overlapping, a total of 66 differentially expressed genes (DEGs) were identified, of which 53 were upregulated and 24 were downregulated." (results 3.1 show 13 downregulated, 53+24 do not add up to 66)

Reviewer #3: Major revision is needed.

(As above stated points are significant to the study design)

**Summary and General Comments**

Reviewer #1: The study is ok; but with no new findings for pathogenesis of deng discovered except cell cycle which is already known, and the microRNA sectis not done rightlyion

Reviewer #2: As stated before, a differentiation in samples from different time (days) since symptoms onset is needed. Since transcriptional profiles are distinct, in order to find better biomarkers it would be required to at least not group convalescent blood samples with initial infection samples. Also, an in vitro experiment in cell lineage (jukarT, B-LCL, ...) in which the identified miRNA were targeted (either via KO by CRISPR/Cas) or maybe the mRNAs were silenced by siRNA would also be interesting. But i also understand that since no funding is linked to this research, the in vitro experiment may be out of question.

Reviewer #3: The study is a good and scientifically sound. Biomarkers are an important biological components in diagnosis and identification of disease and its severity. The DENV infection in tropical and sub-tropical region are big challenges to the health department. Identification of potential biomarkers will help us in diagnosis and identification of drug targets.

The current approach is insilico based analysis hence a clinical validation is needed.

PLOS authors have the option to publish the peer review history of their article (what does this mean?). If published, this will include your full peer review and any attached files.

Reviewer #1: Yes: Mahmoud ElHefnawi

Reviewer #2: No

Reviewer #3: No
---

## [Decision Letter · Decision Letter 1]

8 Jun 2021

Dear Mr. Guo,

Thank you very much for submitting your manuscript "Identification of potential biomarkers in dengue via integrated bioinformatic analysis" for consideration at PLOS Neglected Tropical Diseases. As with all papers reviewed by the journal, your manuscript was reviewed by members of the editorial board and by several independent reviewers. The reviewers appreciated the attention to an important topic. Based on the reviews, we are likely to accept this manuscript for publication, providing that you modify the manuscript according to the review recommendations. 

Sincerely,

Ahmed Mostafa

Associate Editor

A. Desiree LaBeaud

Deputy Editor

Reviewer's Responses to Questions

**Key Review Criteria Required for Acceptance?**

**Methods**

-Are the objectives of the study clearly articulated with a clear testable hypothesis stated?

-Is the study design appropriate to address the stated objectives?

-Is the population clearly described and appropriate for the hypothesis being tested?

-Is the sample size sufficient to ensure adequate power to address the hypothesis being tested?

-Were correct statistical analysis used to support conclusions?

-Are there concerns about ethical or regulatory requirements being met?

Reviewer #2: -Are the objectives of the study clearly articulated with a clear testable hypothesis stated?

Yes. The objective of the study is to identify dengue infection biomarkers. In order to do so the authors analyze public data with a standard methodology.

-Is the study design appropriate to address the stated objectives?

Yes, it is. By analyzing available transcriptomes, filtering DEGs, proceeding with pathway enrichment, protein-protein interaction and then looking for miRNAs, the authors are able to find potential biomarkers for dengue infection. 

-Is the population clearly described and appropriate for the hypothesis being tested? 

Yes, the authors specify all of it in the manuscript. 

-Is the sample size sufficient to ensure adequate power to address the hypothesis being tested? 

Yes, there is enough data from these datasets to account for any statistical analysis.

-Were correct statistical analysis used to support conclusions? 

Yes, the packages and tools the group used are standard and widely accepted in the bioinformatics' field.

-Are there concerns about ethical or regulatory requirements being met? 

No, as they are using public data there is nothing to worry about.

Reviewer #3: As per the authors reply over the reviewer comments, they are acceptable

**Results**

-Does the analysis presented match the analysis plan?

-Are the results clearly and completely presented?

-Are the figures (Tables, Images) of sufficient quality for clarity?

Reviewer #2: -Does the analysis presented match the analysis plan?

Yes, it does.

-Are the results clearly and completely presented?

Yes, they are.

-Are the figures (Tables, Images) of sufficient quality for clarity?

Yes, yes. The .tiff quality is excellent and the figures are disposed in an organized and clear way.

Reviewer #3: Most the author reply is acceptable, except figure 1 explanation

The threshold was set as followed: P<0.05 and |log2FC|≥2. FC: fold change

Review comment by author:

However, considering that it can ensure significant

differences and screen out more differential genes for further analysis and comparability

before data sets, this study took p < 0.05 and FC ≥ 1.5 when analyzing each dataset.

Because when fold change took 2, the number of DEGs obtained from individual data

sets are relatively small.

This both statements contradicts please clarify

Figure 4 and 5 quality is poor

**Conclusions**

-Are the conclusions supported by the data presented?

-Are the limitations of analysis clearly described?

-Do the authors discuss how these data can be helpful to advance our understanding of the topic under study?

-Is public health relevance addressed?

Reviewer #2: -Are the conclusions supported by the data presented?

Yes, they proposed to find biomarkers for dengue infection and so they did.

-Are the limitations of analysis clearly described?

Yes, the authors perfectly describe it in the manuscript. Even though there are no in vitro nor in vivo experiments, and this may limit their findings, this is discussed and accounted for. Nonetheless, these findings are a great contribution for future studies and may be helpful for researches in dengue therapy.

-Do the authors discuss how these data can be helpful to advance our understanding of the topic under study?

Yes, they state it clearly to the readers.

-Is public health relevance addressed?

- Yes, identifying potential biomarkers for disease treatment (and in this case, infection control) are of extreme importance. Even more when talking about dengue, its relevance goes without saying given the number of infected people yearly.

Reviewer #3: The authors reply is acceptable

**Editorial and Data Presentation Modifications?**

Reviewer #2: (No Response)

Reviewer #3: Minor revision

**Summary and General Comments**

Reviewer #2: The authors did a great work with the revision. This sort of bioinformatics analysis is of utmost importance for identifying therapeutic targets and controlling diseases. There is one point I would like to address, though. In your first submission we discussed about different infection's stages and how it may affect the patients' transcriptional profile. I don't know about limma, but when analyzing RNA-seq with DESeq2 there is a test called Likelihood Ratio Test (LRT) that may be used (instead of the default wald test) to identify genes whose expression change overtime. There may be a similar or corresponding test that can be used on limma and may prove to be useful for your future analysis.

Reviewer #3: The authors reply is acceptable

PLOS authors have the option to publish the peer review history of their article (what does this mean?). If published, this will include your full peer review and any attached files.

Reviewer #2: Yes: Igor Salerno Filgueiras

Reviewer #3: No

Figure Files:

Data Requirements:

Reproducibility:

References

---

## [Editor Report · Decision Letter 2]

7 Jul 2021

Dear Mr. Guo,

We are pleased to inform you that your manuscript 'Identification of potential biomarkers in dengue via integrated bioinformatic analysis' has been provisionally accepted for publication in PLOS Neglected Tropical Diseases.

Best regards,

Ahmed Mostafa

Associate Editor

A. Desiree LaBeaud

Deputy Editor

The quality of the figures must be improved "higher resolutions" in the published version of the manuscript

---

## [Editor Report · Acceptance letter]

20 Jul 2021

Dear Mr. Guo,

We are delighted to inform you that your manuscript, "Identification of potential biomarkers in dengue via integrated bioinformatic analysis," has been formally accepted for publication in PLOS Neglected Tropical Diseases.

Best regards,

Shaden Kamhawi

co-Editor-in-Chief

Paul Brindley

co-Editor-in-Chief
